# Serological and Molecular Survey for Dengue Virus Infection in Suspected Febrile Patients in Selected Local Government Areas in Adamawa State, Nigeria

**DOI:** 10.3390/vaccines10091407

**Published:** 2022-08-28

**Authors:** Daniel Thakuma Tizhe, Jacob Kwada Paghi Kwaga, Grace Sabo Nok Kia

**Affiliations:** 1Department of Biochemistry, Ahmadu Bello University, Zaria 810106, Nigeria; 2African Centre of Excellence for Neglected Tropical Diseases and Forensic Biotechnology, Ahmadu Bello University, Zaria 810106, Nigeria; 3Department of Veterinary Public Health and Preventive Medicine, Ahmadu Bello University, Zaria 810106, Nigeria

**Keywords:** *Aedes*, dengue, febrile, malaria, misdiagnosis, serotyping, phylogeny

## Abstract

Dengue is a disease caused by the dengue virus that is primarily transmitted by *Aedes aegypti* mosquitoes. Currently, the disease poses a threat to public health, with about 390 million people reported to be infected annually across the endemic regions of the world. In Nigeria, the disease is under-reported and often misdiagnosed as malaria. This study was designed to conduct a serological and molecular survey for dengue virus infection in febrile patients in three Local Government Areas (LGAs) in Adamawa State, Nigeria, from September through December 2020. Serum samples from 424 patients were analysed by Enzyme-Linked Immunosorbent Assay (CALBIOTECH, Dengue Virus IgM ELISA). Thick and thin smear microscopic techniques were used to determine the presence of malaria parasites. Overall, 19.4% patients were sero-positive for dengue in the three study locations. A total of 11%, 14.5% and 12.3% participants were found to be co-infected with dengue and malaria in Mubi, Jimeta and Numan, respectively. The CDC DENV1-4 RT-PCR Assay reagent was used for serotype-specific detection and identification of circulating serotypes. From the ELISA-positive samples, 11 (2.6%) cases were confirmed to be dengue serotype 1 by Real-Time PCR and sequencing and were found to be in circulation in all the three study areas. With an overall sero-prevalence of 19.4%, dengue virus infection may be one of the major causes of febrile illnesses across the study locations; hence, public healthcare professionals should not neglect other aetiologies of febrile illnesses and the need to conduct laboratory diagnoses to determine the possible causes of febrile illnesses.

## 1. Introduction

Dengue virus is a positive-sense single-stranded RNA virus with a genome size of about 11 kb that belongs to the genus *Flavivirus* and family *Flaviviridae*, transmitted by the *Aedes aegypti* mosquitoes [1]. The virus causes dengue disease, which is endemic in the urban and peri-urban cities across the tropical and subtropical regions of the world and currently poses a major threat to public health [2]. It has been reported that about 390 million cases occur annually worldwide [3,4]. The virus is responsible for more morbidity and mortality compared to other vector-borne viral infectious diseases, with an estimated mortality rate of 24,000 annually [5]. There are four known serotypes in circulation that share antigenic relatedness (DENV1–DENV4), each of which is further divided into specific genotypes. Infection with any one of the serotypes provides a lifelong immunity against the same serotype and partial immunity against the other serotypes [6,7]. Dengue has been classified as dengue with or without warning signs/symptoms and severe dengue [8]. Even though there is no available effective treatment in place, early initiation of rehydration therapy of severe conditions can be successfully managed by careful monitoring and survey of the warning symptoms [9]. For the past ten years, commitments have been made to focus on three ground-laying areas: surveillance for planning and response, reducing the disease burden and consciously changing behaviour patterns to improve vector control [9]. There have been reported cases of dengue outbreaks in Nigeria that may be neglected due to under-reporting, misdiagnosis and lack of awareness [10]. Recent reports indicate that the dengue virus is a major cause of febrile illness in Nigeria, often misdiagnosed as malaria [11]. The majority of public healthcare institutions in Nigeria overlook the possibility of other causes of febrile illness of viral and bacterial origin, putting greater emphasis on the parasitic source [12,13,14]. In Nigeria, the prevalence of dengue disease and circulating viral serotypes are not clear, as the disease is under-diagnosed or misdiagnosed as malaria; misdiagnosis of dengue and other arboviral infections as malaria is probably going to have enormous impacts in the overall management of febrile infection in Nigeria [15]. Data on dengue occurrence and the circulating serotypes are critical for effective control, management, integrated surveillance and outbreak preparedness. Adamawa State was chosen for this study, because the state has not been targeted in previous surveillance studies for dengue in Nigeria. Figure 1 present a map of Adamawa State showing the randomly selected study areas highlighted in colours.

## 2. Materials and Methods

### 2.1. Study Design

This was a cross-sectional study designed to carry out a serological and molecular survey for dengue virus infection in febrile patients in selected LGAs in Adamawa State, Nigeria. Adamawa State is made up of 21 LGAs and is divided into three (3) political zones: northern, central and southern senatorial districts. The three LGAs were randomly chosen by balloting from each zone. From each of the selected LGAs, one secondary and primary healthcare facility each were purposively selected and selected based on convenience. A structured questionnaire was administered to establish an association between behaviour patterns and dengue infection cases. Patients with febrile symptoms consistent with dengue infection such as high body temperature (>38 °C), severe headache, joint and muscular pains, back pain, loss of appetite, tiredness, nausea and rash were recruited. Patients who refused to participate in the study and met the inclusion criteria were excluded. Samples were collected from September–December 2020 through venepuncture by a phlebotomist in all the healthcare facilities.

### 2.2. Ethical Clearance

Ethical clearance was obtained from the Adamawa State Ministry of Health, Ethics Committee (**S/MoH/1131/I**), and clearance from the respective hospitals’ board of ethics (General Hospital and Kolere Primary Healthcare, Mubi; Specialist Hospital and Jimeta Clinic and Maternity, Yola North; General Hospital and Sabon Pege Primary Healthcare, Numan). Written informed consent of adult patients >18 years was obtained. For participants between 13 and 18 years, signed informed consent of both minor and guardian and or parent was obtained. For participants <13 years, signed informed consent was obtained from the parent/guardian.

All experiments and methods were performed in accordance with relevant guidelines and regulations. 

### 2.3. Malaria Diagnosis

Blood samples from recruited participants were collected by venepuncture using 5 mL plain tubes. Malaria parasitaemia was determined by Giemsa-stained thick and thin film microscopy [16].

### 2.4. Dengue Diagnosis

The samples were subjected to centrifugation at 5000× *g* for 5 min. Serum was used to screen for the presence of dengue virus antibody using an Enzyme-Linked Immunosorbent Assay (ELISA)-specific kit for dengue IgM according to the manufacturer’s instructions (CALBIOTECH, Dengue Virus IgM ELISA, DE051M, CA 92020). A positive ELISA result was defined as having an antibody index value >1.1

### 2.5. Viral RNA Extraction

Viral RNA was extracted from human serum (200 µL) using DaAn Gene Co., Ltd. (Guangzhou, Guangdong, China) following the manufacturer’s instructions. A final volume of 60 µL of RNA was obtained and used as a template for cDNA synthesis and for the subsequent Real-Time PCR assay. The concentration (ng/µL) and purity of RNA was determined using Nanodrop spectrophotometer (Thermo Fisher Scientific, Waltham, MA, USA) at 260/280 OD. The extracted RNAs were stored at −80 °C for further analysis.

### 2.6. First Strand cDNA Synthesis

Following RNA extraction, ProtoScript II First Strand cDNA Synthesis (NEB, Hitchin, UK) Kit was used for cDNA synthesis; 2 µL of 50 ng/µL RNA sample was mixed with 2 µL of Oligo d(T)_23_VN to a total volume of 8 µL in nuclease-free water in a sterile microfuge tube and incubated for 5 min at 65 °C. The mixture was briefly spun and promptly put on ice. Then, 10 µL of ProtoScript II reaction mix and 2 µL ProtoScript II enzyme mix were added to reaction tube. The cDNA synthesis reaction was incubated at 42 °C for 1 h. The reaction was inactivated at 80 °C for 5 min and the cDNA products were quantified using Nanodrop spectrophotometer (Thermo Fisher Scientific, Waltham, USA) at 260/280 OD and stored at −20 °C.

### 2.7. Real-Time PCR Serotyping

Dengue virus serotyping was carried using Real-Time PCR (BIO-RAD CFX96 Real-Time System) with CDC DENV1-4 Real-Time RT-PCR assay ancillary reagents. The optimized reaction conditions for PCR amplification involved initial denaturation at 95 °C for 2 min followed by 40 amplification cycles of denaturation and annealing at 95 °C for 15 s and 60 °C for 1 min. Briefly, the CDC DENV1-4 Real-Time RT-PCR Assay includes a set of oligonucleotide serotype-specific primers and dual-labelled hydrolysis (TaqMan^®^) probes for in vitro qualitative detection of DENV serotypes 1–4 from human serum with febrile illness consistent with dengue infection. The targeted regions (DENV1 NS5 gene 112 bp, DENV2 E gene 78 bp, DENV3 prM gene 74 bp, and DENV4 prM gene 89 bp) of the viral genome were amplified by the polymerase chain reaction (PCR). The fluorescently labelled probes anneal to amplified DNA fragments and the fluorescent signal intensity was monitored by a BIO-RAD CFX 96 instrument during each PCR cycle. Target amplification was recorded as an increase and accumulation of fluorescence over time in contrast to the background signal. The assay was performed in a multiplex reaction targeting each DENV serotype with a different coloured probe, i.e., each Taqman probe targets a single DENV serotype and is conjugated to a fluorophore that emits fluorescence at different excitation wavelengths (5′-FAM DENV-1, 5′-HEX DENV-2, 5′-Texas Red DENV-3 and 5′-Cy5 DENV-4). According to the CDC recommendation, a specimen is considered positive if the DENV marker amplification curve crosses the threshold line within 37 cycles (<37 Ct) [17]. 

Serotyping through identification and amplification of the nonstructural protein gene (NS5), envelope protein (E) gene and membrane protein gene was performed by the CDC DENV1-4 RT-PCR specific primer sequences (5′–3′), as presented in Table 1. 

### 2.8. Sequencing and Phylogenetic Analysis

The DENV1 nonstructural partial gene (NS5 <1…109>) region was amplified using a two-step RT-PCR (ProtoScript, NEB, UK) kit with the CDC DENV1-F and DENV1-R primers (CAAAAGGAAGTCGTGCAATA and CTGAGTGAATTCTCTCTACTGAAC). The PCR purified products were sequenced by Inqaba Biotec West Africa Ltd. (Ibadan, Nigeria) using the DENV1-F and DENV1-R primers.

The sequence chromatogram files were edited for bad calls using BioEdit. The sequences were compared with available sequences using the Basic Local Alignment Search Tool (nBLAST) and the GenBank database to validate the identity of the viral serotype.

The sequences were aligned using MUSCLE and Molecular Evolutionary Genetics Analysis Software (MEGA X) version 10.1.7 was used for phylogenetic analysis using the Maximum Likelihood statistical method. The robustness of the topology tree was evaluated during 1000 bootstrap replicates based on the Tamura-Nei model.

### 2.9. Data Analysis

Statistical Package for Social Science (SPSS) version 24 was employed in the analysis of data. The Chi-squared test was used to search for association between dengue virus infection and the socio-demographic information of the participants in the study locations. Level of significance was set at *p* ≤ 0.05. The degree of association between dengue infection and behaviour patterns was examined by calculating the odds ratio framed at 95% confidence intervals. The Maximum Likelihood statistical algorithm evaluated with 1000 bootstrap replicates based on the Tamura–Nei model was used in the phylogenetic analysis.

## 3. Results

Out of the 424 samples collected, 146 (34.4%) were collected from Mubi, 124 (29.4%) from Jimeta, Yola, and 154 (36.2%) from Numan. An overall total of 82 (19.3%) patients were positive for dengue IgM from the three study locations, with 11 positive cases confirmed to be DENV1 by RT-PCR and sequencing. Among the 82 patients, 50 (11.8%) were females and 32 (7.5%) were males. Of the 82 patients, 53 (12.5%) were found to be co-infected with dengue and malaria. Out of the 424 recruited participants, 341 (80.4%) admitted to having been exposed to mosquito bites in their homes and 82 (19.3%) were positive for dengue IgM. A total of 203 (47.9%) out of the recruited participants admitted to having some form of open water in or around their surroundings and 71 (16.7%) were positive for dengue IgM detection assay. A significant statistical association at *p*-value (0.01, 0.02 and 0.01) was observed between the behaviour patterns and dengue infection across the three study locations.

Following Real-Time PCR assay, amplicons from the 11 confirmed cases were subjected to sequencing, and the assembled sequences of the NS5 partial gene (DENV1) were deposited in the DNA Data Bank of Japan under accession number LC702703. 

Table 2 presents the dengue virus antibody (IgM) detection assay using ELISA for samples collected from Mubi, Jimeta and Numan. Twenty-nine (19.9%) participants from Mubi were positive for dengue IgM antibody detection. Twenty-four (19.4%) participants from Jimeta were positive for dengue IgM antibody detection. Twenty-nine (18.8%) participants from Numan were positive for dengue IgM antibody detection. The age-specific range of dengue sero-prevalence was predominant in the age group 0–35 years, as presented in Table 3, Table 4 and Table 5.

Table 6 presents data on dengue and malaria co-infection for samples collected from Mubi, Jimeta and Numan. Out of the 146 samples tested from Mubi, 16 (11%) were found to be co-infected with dengue and malaria. Out of the 124 samples tested from Jimeta, 18 (14.5%) were found to be co-infected with dengue and malaria. Out of the 154 samples tested from Numan, 19 (12.3%) were found to be co-infected with dengue and malaria.

Figure 2, Figure 3 and Figure 4 are bar charts showing participants that were exposed to mosquito bites without control measures in place in Mubi, Jimeta and Numan. Out of the 146 participants in Mubi, 118 (80.8%) admitted to having been exposed to mosquito bites in their homes, and 29 (19.9%) of them were positive for dengue IgM. Out of the 124 participants in Jimeta, 107 (86.3%) admitted to having been exposed to mosquito bites in their homes, and 24 (19.4%) of them were positive for dengue IgM. Out of the 154 participants in Numan, 116 (75.3%) admitted to having been exposed to mosquito bites, and 29 of them (18.8%) were positive for dengue IgM.

Figure 5, Figure 6 and Figure 7 present participants who had some form of open water around their dwelling places in Mubi, Jimeta and Numan. Among the 146 participants in Mubi, 73 (50.0%) had some form of open water in or around their surroundings, and 26 (17.80%) of them were positive for dengue IgM. Among the 124 participants in Jimeta, 55 (44.4%) had some form of open water in or around their surroundings, and 21 (16.9%) of them were positive for dengue IgM. Among the 154 participants in Numan, 75 (48.7%) had some form of open water in or around their surroundings, and 24 (15.6%) of them were positive for dengue IgM.

Of the 82 ELISA-positive results across the study areas, 11 were confirmed to be DENV1 by Real-Time PCR assay and sequencing and grouped as genotype V.

Figure 8 presents the inferred phylogenetic tree of DENV1 using the Maximum Likelihood statistical method. The robustness of the tree topology was evaluated during a 1000 bootstrap non-parametric analysis based on the Tamura–Nei model.

## 4. Discussion

Dengue is a major arthropod-borne viral disease of humans, and its control and management remain the primary priority of public health for many endemic countries. Unlike the situation in the Asia-Pacific and Latin American regions, where dengue burden and epidemiology are well documented, the burden and epidemiology of dengue virus infection in Africa including Nigeria is not clear, even though there are reported cases [11].

In an attempt to improve surveillance and to track the viral burden and epidemiology in Nigeria, this population-based survey was conducted in secondary and primary healthcare facilities to determine the occurrence of dengue virus infection and serotypes in Mubi, Yola North (Jimeta) and Numan LGAs in Adamawa State, Nigeria. Febrile patients with clinical signs and symptoms compatible with dengue virus infection, which were referred to a laboratory for malaria parasitaemia diagnosis and who consented to participate before sample collection, were recruited into this study. Despite being undetected prior to this study, active transmission of dengue virus, dengue and malaria co-infection were detected across the three study locations. An overall sero-prevalence of 19.4% for dengue virus infection was established.

In Mubi, out of the 146 patients, 97 (66.4%) tested positive for malaria parasites and 49 (33.6%) were negative. Of the 29 ELISA-positive results, 3 samples were confirmed to be DENV1 by RT-PCR and sequencing. The age-specific range of dengue sero-prevalence was observed to be significant in the younger participants aged 0–35 years with a *p*-value of 0.003. This could be plausible, since these age groups are known to be actively involved in outdoor activities, where possible contact with a vector is very likely. A sero-prevalence of 19.9% for dengue virus IgM was obtained in Mubi. This, therefore, means that younger people within the age range of 5–35 years should be the target age for the dengue vaccination programme, if the need becomes imperative. Out of the 146 patients, 16 (11%) were found to be co-infected with dengue and malaria. This observation has brought to light the fact that examination of febrile illnesses should not be limited to parasites and bacteria, but that flaviviruses can equally cause febrile illnesses.

In Jimeta, out of the 124 patients, 83 (66.9%) tested positive for malaria parasites and 41 (33.1%) were negative. Of the 24 ELISA-positive results, 3 samples were confirmed to be DENV1 by RT-PCR and sequencing.

Out of the 124 patients, 18 (14.5%) were found to be co-infected with dengue and malaria. This again suggests that febrile illnesses should not be limited to malaria and typhoid, but that flavivirus infection can result in febrile illness; hence, proper laboratory diagnosis for febrile illnesses should be readily available and recommended before drug prescription.

In Numan, out of the 154 patients, 97 (63.0%) tested positive for malaria parasites and 57 (37.0%) were negative. Of the 29 ELISA-positive results, 5 samples were confirmed to be DENV1 by RT-PCR and sequencing. Like Jimeta, dengue sero-prevalence was found across all the age groups but was more pronounced in the younger age groups. Out of 154 patients, 19 (12.3%) were found to be co-infected with dengue and malaria. This further confirms that flaviviruses are capable of causing febrile illness, as it was found to be present across all the study locations.

From the findings, malaria may not be the only mosquito-borne disease causing febrile illness across the study locations. With dengue sero-prevalence rates of 19.9%, 19.4% and 18.8% in the three study locations, there may be an active transmission of dengue across the entire state and possibly beyond, due to the fact that dengue is frequently being transported from one place to another by infected travellers coupled with inherent behaviour patterns of the population that encourage vector breeding. The development implies that dengue may be endemic in these areas, but undetected or possibly misdiagnosed as malaria.

The co-infection of febrile patients with dengue and malaria in the study could suggest the environmental suitability for *Anopheles* and *Aedes* mosquitoes across these locations. A similar scenario of dengue and malaria co-infection was reported by Ayukekbong [15] in Ibadan, where 10% of malaria patients had active dengue infection.

In a recent serological survey of dengue IgM, Oladipo et al. [18] reported a 17.2% prevalence in seemingly healthy individuals in Ogbomosho, while Adesina and Adeniji [19] reported a sero-prevalence of dengue IgM of 25.7% among febrile volunteer subjects in Ile-Ife, western Nigeria.

In Jos and Ibadan, 2.2% and 35% of febrile subjects were reported to be positive for the dengue nonstructural protein one (NS1) antigen, respectively [11]. In another separate study, Hamisu et al. [20] reported that 37.4% of subjects were positive for dengue IgM in Maiduguri, and Bello et al. [21], in a study conducted in Kaduna, reported 51.1% and 48.9% positive for dengue IgM in females and males, respectively.

Only dengue virus serotype 1 was identified in the three study locations, and this may highlight the active occurrence of dengue virus in the study areas. In a similar study in Ibadan, DENV serotypes 1 and 2 were detected, while DENV serotype 1 was reported in Abeakuta [22]. In Lagos, Ayolabi et al. [23] reported the circulation of DENV serotypes 1 and 3 in febrile patients. In a study conducted by Yousseu et al. [24] in Douala, Cameroon, active circulation of DENV serotype 1 was reported. Parreira et al. [25] reported DENV serotype 1 to be responsible for the outbreak in Angola. However, other reports indicated that DENV serotypes 2 and 3 may cause more severe forms of dengue infection than the other serotypes [26].

The detection of DENV serotype 1 in this study could explain why severe cases of dengue virus infection due to secondary infection and cross-reactivity with other serotypes have not been reported in the three study areas. However, this does not rule out potential outbreaks and may not be the current situation, as possible cases of importation of other serotypes cannot be ruled out.

Currently, dengue prevention and control depend heavily on effective vector control measures; hence, sustained local involvement can improve vector control efforts substantially, as local risk to dengue outbreaks is linked to the population’s knowledge, attitude as well as behavioural practices that encourage vector breeding. Communities are encouraged to avoid exposure to mosquito bites especially during the day time, rid their environment of possible mosquito breeding sites (as proximity of breeding sites is a significant risk factor to dengue transmission) and remain active in community engagement through sensitization and mobilization against mosquito-borne diseases.

## 5. Conclusions

This study established the occurrence of dengue virus infection in human subjects tested in Mubi, Jimeta and Numan LGAs in Adamawa State, Nigeria. Sero-prevalence rates of 19.9%, 19.4% and 18.8% were generated, respectively. Dengue and malaria co-infection was established: this suggests that febrile illnesses are not only limited to malaria and typhoid. A significant association was established between behaviour patterns and dengue infection. As at the time this study was conducted, DENV serotype 1 was found to be in circulation across the three study locations; therefore, the public healthcare professionals should not neglect other aetiologies of febrile illnesses and the need to conduct laboratory diagnoses to determine the possible causes of infection.

## Figures and Tables

**Figure 1 vaccines-10-01407-f001:**
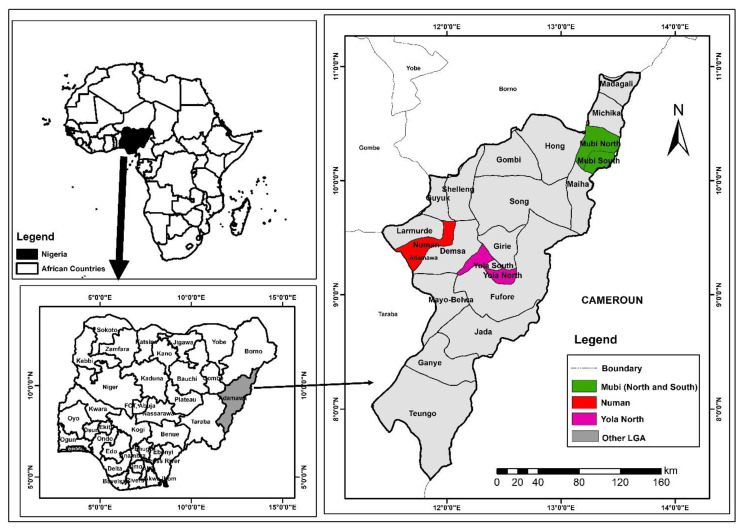
A map of Adamawa State showing the study areas.

**Figure 2 vaccines-10-01407-f002:**
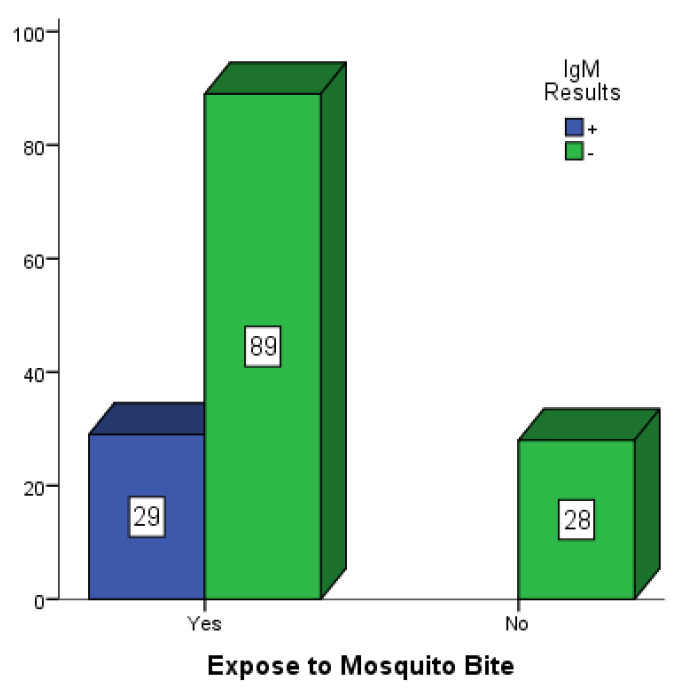
Participants exposed to mosquito bites in Mubi.

**Figure 3 vaccines-10-01407-f003:**
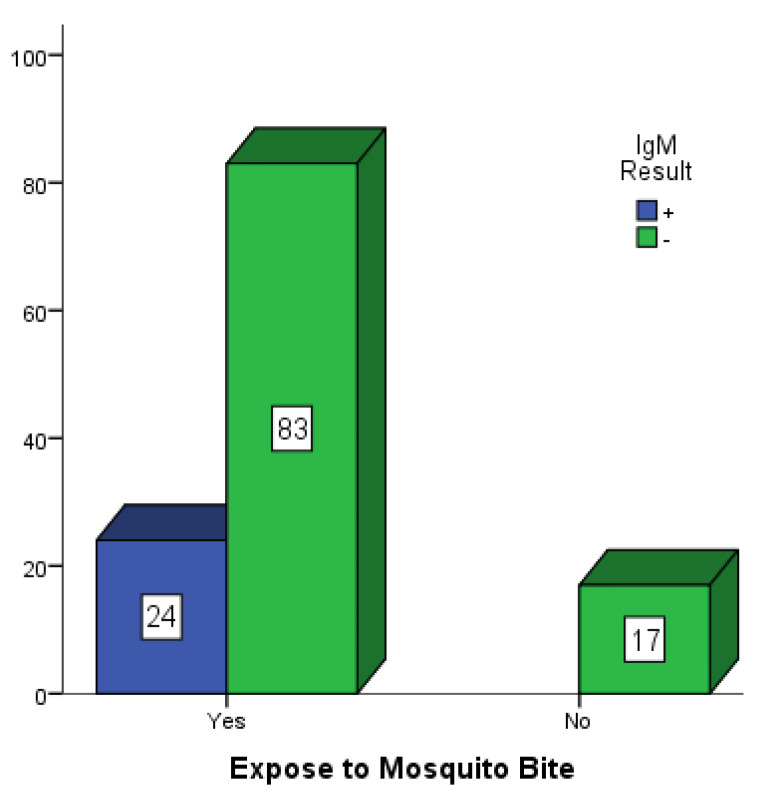
Participants exposed to mosquito bites in Jimeta.

**Figure 4 vaccines-10-01407-f004:**
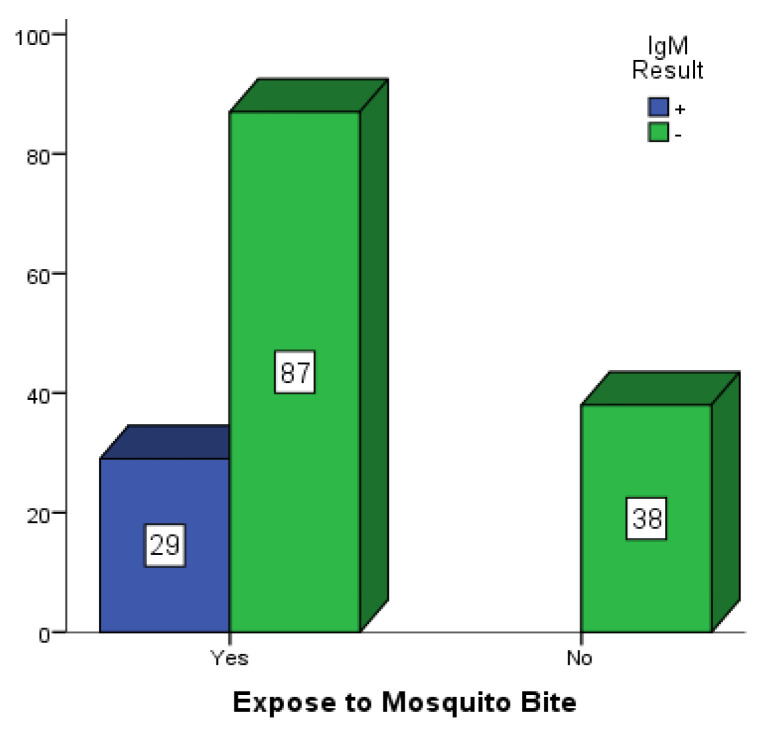
Participants exposed to mosquito bites in Numan.

**Figure 5 vaccines-10-01407-f005:**
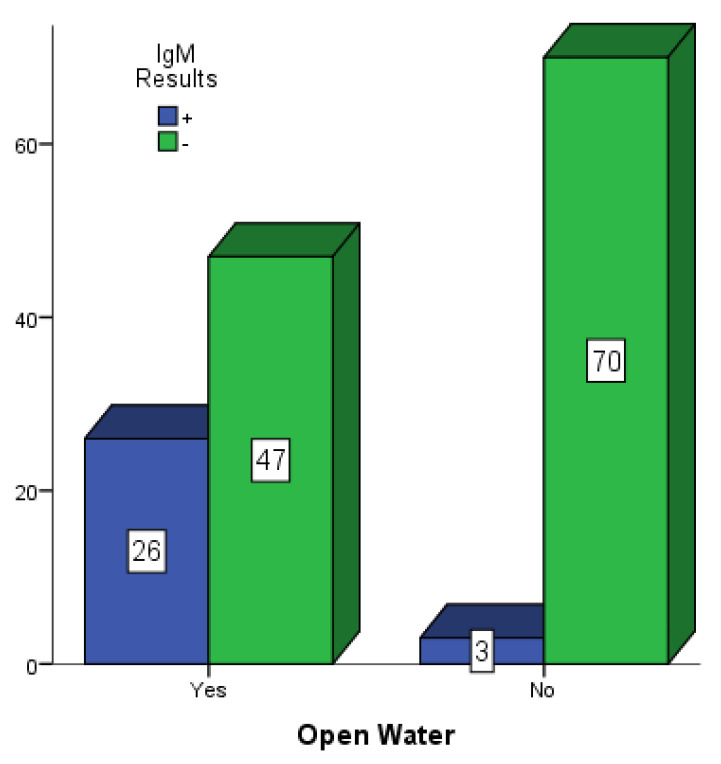
Participants with open water in their surroundings in Mubi.

**Figure 6 vaccines-10-01407-f006:**
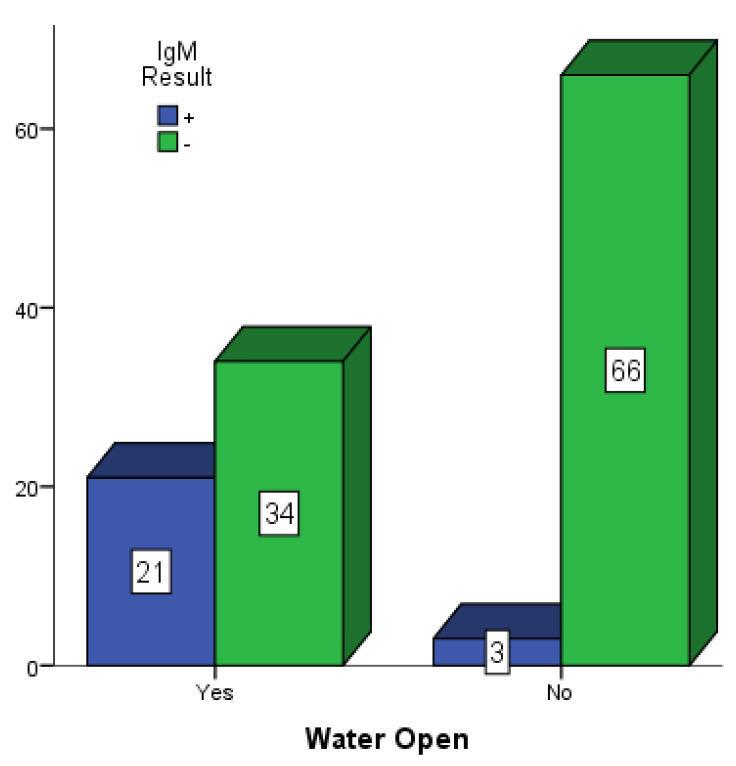
Participants with open water in their surroundings in Jimeta.

**Figure 7 vaccines-10-01407-f007:**
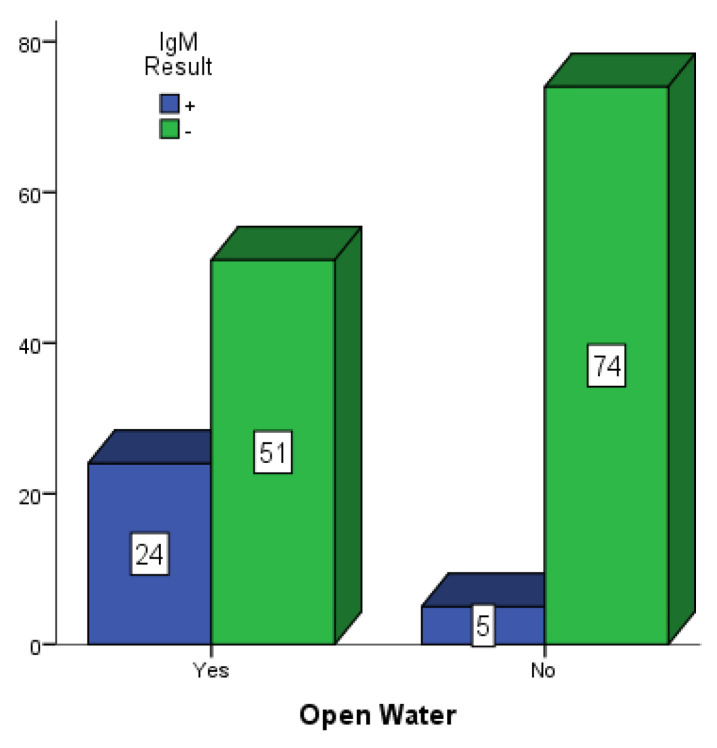
Participants with open water in their surroundings in Numan.

**Figure 8 vaccines-10-01407-f008:**
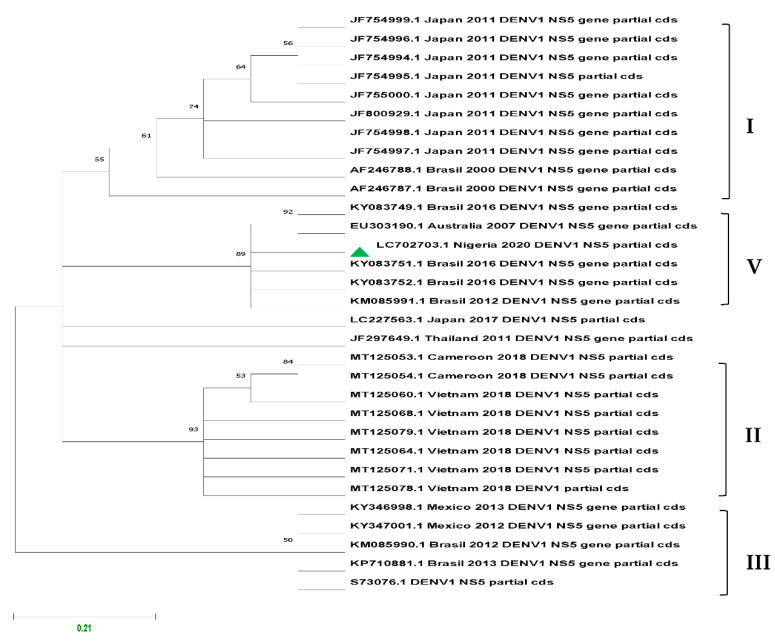
Maximum Likelihood phylogenetic tree of DENV1 based on partial NS5 gene of 109 nt length and 31 sequences deposited in GenBank generated using GTR + G nucleotide substitution model. Bootstrap values are shown on branch nodes and annotation on the right denote genotypes (Asian I, Asian/American II, Cosmopolitan III, African/American V). Sequence from this study is indicated with green marker with accession number LC702703.1.

**Table 1 vaccines-10-01407-t001:** CDC DENV1-4 RT-PCR primer sequences (5′–3′).

DENV1F	CAAAAGGAAGTCGTGCAATA
DENV1R	CTGAGTGAATTCTCTCTACTGAAC
DENV2F	CAGGTTATGGCACTGTCACGAT
DENV2R	CCATCTGCAGCAACACCATCTC
DENV3F	GGACTGGACACACGCACTCA
DENV3R	CATGTCTCTACCTTCTCGACTTGTCT
DENV4F	TTGTCCTAATGATGCTGGTCG
DENV4R	TCCACCTGAGACTCCTTCCA

**Table 2 vaccines-10-01407-t002:** ELISA detection of dengue IgM.

Areas	IgM	Frequency	Percent
Mubi	+	29	19.9
Jimeta	+	24	19.4
Numan	+	29	18.8
Total		82	19.4

**Table 3 vaccines-10-01407-t003:** Dengue virus infection detection among patients in Mubi by age group.

Age Groups	IgM Results (+)	IgM Results (−)	Total Patients	Age Group Sero-Prevalence (%)	*p*-Value
0–5	1	25	26	0.68	
6–10	9	9	18	6.16	
11–15	2	12	14	1.37	
16–20	10	17	27	6.85	
21–25	0	10	10	0.00	
26–30	3	11	14	2.05	
31–35	2	9	11	1.37	
36–40	0	7	7	0.00	
41–45	2	7	9	1.37	
46–50	0	1	1	0.00	
51–55	0	3	3	0.00	
>56	0	6	6	0.00	
Total	29	117	146	19.9	0.003

**Table 4 vaccines-10-01407-t004:** Dengue virus infection detection among patients in Jimeta, Yola, by age group.

Age Groups	IgM Results (+)	IgM Results (−)	Total Patients	Age Group Sero-Prevalence (%)	*p*-Value
0–5	2	6	8	1.61	
6–10	3	7	10	2.42	
11–15	4	9	13	3.23	
16–20	1	18	19	0.81	
21–25	1	12	13	0.81	
26–30	9	16	25	7.26	
31–35	1	14	15	0.81	
36–40	1	5	6	0.81	
41–45	0	7	7	0.00	
46–50	1	3	4	0.81	
51–55	0	1	1	0.00	
>56	1	2	3	0.81	
Total	24	100	124	19.4	0.238

**Table 5 vaccines-10-01407-t005:** Dengue virus infection detection among patients in Numan by age group.

Age Groups	IgM Results (+)	IgM Results (−)	Total Patients	Age Group Sero-Prevalence (%)	*p*-Value
0–5	1	15	16	0.65	
6–10	4	8	12	2.60	
11–15	5	13	18	3.25	
16–20	6	19	25	3.90	
21–25	4	12	16	2.60	
26–30	2	12	14	1.30	
31–35	1	8	9	0.65	
36–40	1	9	10	0.65	
41–45	3	4	7	1.95	
46–50	1	9	10	0.65	
51–55	0	5	5	0.00	
>56	1	11	12	0.65	
Total	29	125	154	18.8	0.417

**Table 6 vaccines-10-01407-t006:** Dengue and malaria co-infection.

Areas	Co-Infection	Frequency	Percent
Mubi	Yes	16	11.0
Jimeta	Yes	18	14.5
Numan	Yes	19	12.3
Total		53	17.7

## Data Availability

Data supporting this article are available in the DDBJ data repository (https://getentry.ddbj.nig.ac.jp/getentry/na/LC702703/) (accessed on 23 March 2022).

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
