# Peer review of "Serological and Molecular Survey for Dengue Virus Infection in Suspected Febrile Patients in Selected Local Government Areas in Adamawa State, Nigeria"

_vaccines, 2022, doi:10.3390/vaccines10091407_

Round 1
Reviewer 1 Report
The authors present a serological and molecular survey for dengue virus infection in suspected malaria patients from selected areas of Adamawa State, Nigeria. This is an important study that highlights the ongoing circulation of dengue virus in the area, and the potential misdiagnosis of patients as malaria when other potential causes of febrile illness are not considered and investigated. Apart from my only major concern about the fit between the manuscript and the scope of the specific journal it was submitted to, I feel the manuscript makes a meaningful contribution to understanding the underlying burden of dengue in Nigeria, and it will hopefully contribute towards public health interventions and strengthen vector control efforts.
I have the following specific comments/suggestions for the authors to improve their manuscript:
- The authors describe the approach followed to select specific LGAs in the Adamawa State of Nigeria for this study, however there is no explanation as to why this specific state was the target of this study. Is it because this specific state has not been targeted in previous surveillance studies for dengue in Nigeria, or was it based on convenience? Whatever the reason, the authors should dedicate a short paragraph in the introduction to highlight the importance of this state specifically and why it was chosen for the study.
- The authors refer to the 11 RT-PCR positive DEN-1 cases as "isolates". However, this infers that virus was actually isolated from these cases. The authors should change the way they refer to these 11 cases throughout the text. For example in line 154-156, the authors could rather state "... with 11 positive cases confirmed to be DENV1 by RT-PCR and sequencing." Etc.
- Line 176, the authors should state that the data mentioned in this specific paragraph refers to results from Mubi patients specifically.
- Figures 1.1 to 1.7: why are the figures numbered like this? Why not just number them Figure 1 to Figure 7?
- Suggestion: Figure 1.1, 1.3 and 1.5 could be one composite figure so that readers can look at the proportion of participants exposed to mosquito bites in all three LGAs in one figure, which will make for easier comparison
- Likewise, Figure 1.2, 1.4 and 1.6 could also be one composite figure for easier comparison of participants with open water in their surroundings in the three LGAs
- Line 196-197: The age specific range seems to be predominant in the age group 0 - 30, rather than 0 - 35 years as mentioned in the text. However, considering that this is not supported by the statistics anyway (p = 0.238), this should technically not be mentioned here, or should come with a statement that this observed predominance is not statistically significant (similar to how it is described in the Discussion section)
- Figure 1.7 (phylogenetic tree): the visual aspect of the tree can be improved by more clearly highlighting the Nigerian sequences, either using a symbol next to the placement in the tree, and/or using larger font or bold text
- Line 272 states that seroprevalence was higher in the 6-30 year age range, but in the results section it stated that this age range was 6-20 (line 171). Please keep it consistent.
- It is very obvious when looking at the rate of occurrence in the three LGAs that they are all very similar, both for dengue (19.9%, 19.4% and 18.8%) and malaria (66.4%, 66.9% and 63.0%). This is interesting and suggests that the epidemiology in these three areas are very similar for both diseases. It would be great if the authors could include a map (at least of the state), to show the location of these LGAs in the state; this will be helpful to readers who are not familiar with Nigeria.
- Related to the above comment, the statement by the authors in line 316 about the difference of dengue occurrence noted in the LGAs is not valid, and should not be mentioned; particularly because the authors later state that this "difference" is not statistically significant anyway. The same small difference is evident in the malaria occurrence.
Author Response
The comments and suggestions made by the reviewers has been properly addressed and adopted.
Reviewer 2 Report
This is a very preliminary study to be published and authors need to do more testing (real time PCR in all cases) to find the prevalence of dengue among febrile illness cases. The number of cases positive by IgM ELISA has been given as table and graphs from three locations and may be clubbed together.
Line 38 immunological serotypes –Change it to serotypes
Lines 39-41 : Infection with any one of the serotypes which are usually asymptomatic with mild manifestations provides lifelong immunity against the same serotype and partial immunity against the other serotypes – The sentence is not clear. Needs modification
Section 2.2 should come last in the methods
The authors have performed real time PCR only in IgM positive cases. All Febrile illness cases with post onset day of symptoms less than 5 days should be checked by real time PCR.
Since IgM can persist for three to five months, cases positive for malaria by smear microscopy and dengue IgM cannot be considered as coinfection cases. To be considered as coinfection, they should be positive for malaria by smear microscopy and dengue real time PCR
Phylogenetic analysis should be done in a detailed way and should include reference sequence for different dengue serotype I genotypes and the authors should be able to find out the genotype of circulating virus.
Author Response
The comments and suggestions made by the reviewer has been addressed and submitted as a word document.

Round 2
Reviewer 2 Report
This again is a very preliminary study to be published in Vaccines
The response that authors have provided with regard to phylogenetic analysis is not sufficient. All possible data available from study need to be extracted such as genotype of the dengue virus serotype and this is possible by proper phylogenetic analysis, Even though the sequenced region is small, still the authors should attempt extract all possible details.
Authors have mentioned about sequencing. But no information about which region and which gene was sequenced. Detailed methodology of sequencing with primer sequence need to be provided.
Author Response
The authors were able to provide some useful information as regards the dengue serotype and genotype in circulation in the study areas.
The gene sequence target for serotype 1 according to the CDC DENV1-4 RT-PCR assay is NS5 partial CDs (<1....109>). The primers sequence used in this study were provided by the CDC. The primers sequence are provided in the manuscript.
More information about the sequencing method has also been provided.
Round 3
Reviewer 2 Report
I am happy that the authors have performed phylogenetic analysis and reported that the dengue virus 1 serotype belongs to genotype V.
My concerns are
In the phylogenetic genetic tree, other genotypes also need to be indicated.
The authors have got only one sequence. This need to be indicated in the text. Using one sequence they are claiming that genotype V is circulating in all locations which need corrections (line 362).
Author Response
We want to express our sincere gratitude to the reviewers for their comments and suggestions to improve our manuscript.
As suggested, the genotypes of other DENV1 serotype has been indicated. The sequence from the current study has equally been indicated with a green marker with accession number LC702703.1.